# Staging Investigations in Asymptomatic Early Breast Cancer Patients at the Cancer Centre of Southeastern Ontario

**Dalia Kamel** [1,2,*], **Veronica Youssef** [1,3], **Wilma M. Hopman** [1] **and Mihaela Mates** [1]

1   Cancer Center Southeastern Ontario, Queen's University, Kingston, ON K7L 3N6, Canada;
    veronicayoussef12@gmail.com (V.Y.); Wilma.Hopman@kingstonhsc.ca (W.M.H.);
    Mihaela.Mates@kingstonhsc.ca (M.M.)
2   Medical Oncology Department, Mater Misericordiae University Hospital, Eccles Street,
    D07 R2WY Dublin 7, Ireland
3   Faculty of Medicine and Health Sciences, McGill University, Montreal, QC H3G 2M1, Canada
*   Correspondence: dsskamel@hotmail.com

**Abstract:** Background: In 2012, the American Society for Clinical Oncology (ASCO) identified five key opportunities in oncology to improve patient care, recommending against imaging tests for the staging of patients with early breast cancer (EBC) at low risk for metastases. Similarly, the European Society of Medical Oncology (ESMO) guideline does not support radiological staging in asymptomatic EBC (aEBC). The purpose of this study was to assess local practice and outcomes of staging investigations (SIs) in aEBC at the Cancer Centre of Southeastern Ontario (CCSEO). Methods: A retrospective electronic and paper chart review was undertaken to identify all aEBC patients treated at our institution between January 2012 and December 2014. Patients with pathological staging of T1-T2 and N0-1 with any receptor status were included. We collected patient demographics, treatment and pathologic tumor characteristics. The use and outcomes of initial and follow-up SIs were recorded. Data were analyzed to determine associations between the use of SIs and clinical characteristics (chi-square tests, independent samples t-tests and Mann–Whitney U tests). Results: From 2012 to 2014, 295 asymptomatic EBC patients were identified. The mean age was 64, 81% were postmenopausal and 76% had breast conserving surgery. Stage distribution was as follows: stage I 42%, stage IIA 37% and stage IIB 21%. Receptor status was as follows: ER+ 84%, HER2+ 13% and triple negative 12%. Adjuvant chemotherapy was received by 36%, Trastuzumab by 10% and endocrine therapy by 76% of patients. Baseline SIs were performed in 168 patients (57%) for a total of 332 tests. Overt metastatic disease was found in five patients (one bone scan and four CT scans). Seventy-one out of the 168 patients (42%) who received initial staging imaging underwent 138 follow-up imaging tests, none of which were diagnostic for metastases. Nine patients with suspicious CT findings underwent biopsies, of which four were malignant (one metastatic breast cancer and three new primaries). Factors significantly associated with SI were as follows: younger age ($p = 0.001$), premenopausal status ($p = 0.01$), T2 stage ($p < 0.001$), N1 stage ($p < 0.001$), HER2 positive ($p < 0.001$), triple negative status ($p = 0.007$) and use of adjuvant chemotherapy ($p < 0.001$). Conclusions: Over a 3-year period at our institution, more than 50% of aEBC patients underwent a total of 470 initial and follow-up staging tests, yielding a cancer diagnosis (metastatic breast cancer or second primary cancer) in four patients. We, therefore, conclude that routine-staging investigations in aEBC patients have low diagnostic value, supporting current guidelines that recommend against the routine use of SI in this population.

**Keywords:** early breast cancer; staging investigations

## 1. Introduction

In recent decades, routine radiological staging for newly diagnosed breast cancer has been employed to detect occult metastases and for patient reassurance [1]. It also served as baseline investigations for follow up and for comparison with similar scans from other

centers. This practice was supported by several remote studies that showed a relatively high rate of metastatic bone disease detected with baseline and follow-up bone scans in histological stage I and II breast cancer [2,3]. For example, in one study [2], 18% of stage I and 41% of stage II cancers had positive bone scans confirmed by follow-up studies.

Since then, it has become apparent that the yield of staging tests is much lower; however, the practice has persisted [4,5].

In 2000, Cancer Care Ontario—Program in Evidence-based Care—published a systematic review on the detection rate of metastases through staging investigations, and it showed a low incidence of metastatic disease in asymptomatic patients. Metastatic bone disease was detected by bone scans in 0.5%, 2.4% and 8.3% of patients with pathological stage I, II and III breast cancer, respectively. To a lesser extent, liver metastases were detected by ultrasound in 0%, 0.4% and 2% of patients with pathological stage I, II and III breast cancer, respectively [6]. It is likely that positive findings decreased over time with the improvement in the resolution of imaging.

Based on this systematic review, Cancer Care Ontario (CCO) endorsed the recommendation for routine postoperative bone scan for pathological stage II and III breast cancer and liver ultrasonography and chest radiography for pathological stage III breast cancer. Chest X-ray has lower sensitivity compared to thoracic computer tomography (CT) in the detection of abnormalities within lung parenchyma, lymphadenopathy and axial skeleton [7,8].

Similarly, liver ultrasound is less sensitive than CT abdomen in detecting liver or other intra-abdominal metastases; however, it remains a less expensive method for staging with avoidance of radiation. In current practice, thoracic and abdominal CT scans have largely replaced the use of chest X-ray and liver ultrasound for staging purposes [9].

In April 2012, the American Society for Clinical Oncology (ASCO) identified five key opportunities in Oncology to improve patient care through the Choosing Wisely initiative [10]. The authors highlight that patient care costs are rising, due in part to the unnecessary use of health care resources. After careful consideration by experienced oncologists, five practices are highlighted that are in common use, despite the absence of evidence supporting their clinical value. Specifically for breast cancer, there is no evidence demonstrating benefits for the use of PET, CT or radionuclide bone scans in asymptomatic individuals with newly diagnosed clinical stage I or II disease. Additionally, unnecessary imaging can lead to harm through unnecessary invasive procedures, overtreatment, unnecessary radiation exposure and misdiagnosis. The recommendation of this guideline is to not perform staging scans in patients with early breast cancer at low risk for metastasis. Similarly, the European Society of Medical Oncology (ESMO) guideline does not support radiological staging in asymptomatic EBC (aEBC) [11]. The updated CCO guideline [12] and ASCO Choosing Wisely campaign [13] continue to recommend against routine staging in patients without symptoms.

This study aims to assess the local practice and outcome of SIs in aEBC patients treated at our institution. In the context of the ASCO Choosing Wisely campaign recommendation, the frequency of SI reflects on the adoption of the guideline in standard practice, and we anticipate low frequency of SI if adherence to the guideline is high.

## 2. Patients and Methods

### 2.1. Population and Data Collection

Medical and electronic charts records were reviewed for all patients with pathological stage T1-T2, N0/N1 breast cancer treated at the Cancer Centre of Southeastern Ontario (CCSEO) between January 2012 and December 2014. The definition of early breast cancer is not uniform across centers globally. Some guidelines include breast tumors more than 5 cm in size in the absence of regional lymphadenopathy (stage IIB, T3N0M0) as locally advanced breast cancer. For consistency, we excluded T3N0 tumors from this early breast cancer cohort. Asymptomatic patients were considered those with a negative review of systems documented in the chart. We excluded patients with prior ipsilateral breast cancer

(n = 27), a second active cancer (n = 6) and patients with symptoms suggestive of metastatic disease at the time of diagnosis (n = 2). We collected basic demographic information, medical history, mode of detection, tumor and treatment characteristics, the type of SI and their outcome, as well as follow-up investigations. The age-adjusted Charlson Comorbidity Index (CCI) was used for the assessment of comorbidities with a baseline of 2, as all patients had histologically proven breast cancer.

Ethics approval for this study was obtained from Queen's University Research Ethics Board.

### 2.2. Statistical Analysis

Data were compiled in a Microsoft Excel file and imported into IBM SPSS, version 26.0, for statistical analysis. Data were analyzed descriptively, including means and standard deviations for continuous data, and frequencies and percentages for categorical data. The Shapiro–Wilk test was used to assess the underlying distribution for continuous data. Analysis was performed for the entire sample or subsets, as appropriate. Chi-square tests or the Fisher's Exact test in the event of small cell sizes, independent samples t-tests and Mann–Whitney U tests were used to determine associations between the use of SI and clinical characteristics. A *p*-value of <0.05 was used to define statistical significance, and no adjustments were made for multiple comparisons.

## 3. Results

### 3.1. Patient Demographics and Treatment Characteristics

We identified a total of 295 eligible asymptomatic patients from January 2012 to December 2014. The mean age was 63.6 years (range 30–92); most were postmenopausal (81%), with age-adjusted CCI of more than 4 (79%) and detected by screening mammography most commonly (n = 144; 49%). Tumor characteristics were as follows: stage I 42%, stage IIA 37% and stage IIB 21%. The receptor status was hormone receptor (HR) positive in 84% of patients (defined as estrogen (ER) and/or progesterone (PR) positive), human epidermal growth factor receptor 2 (HER 2)-positive disease in 13% and triple negative (ER/PR/HER2 negative) in 12% of patients, respectively. The majority of patients (n = 188; 64%) did not receive adjuvant chemotherapy due to low clinico-pathologic risk (n = 62; 21%), an Oncotype DX score of less than 18 (n = 44; 15%), patient preference (n = 40; 14%) and co-morbidities (n = 42; 14%). Of the 249 (84%) patients with HR-positive breast cancer, 24 (10%) did not receive endocrine therapy (not recommended in nine patients, and 15 patients declined). Seven patients with HER2-positive disease did not receive Trastuzumab for the following reasons: not recommended (n = 1), patient preference (n = 2) and co-morbidities (n = 4). (See Table 1 for full details).

### 3.2. Baseline SI and Follow-Up Investigations (FUI)

Baseline radiological staging was performed in 168 (57%) patients with a total of 332 scans performed as follows: CT scans (n = 152), bone scans (n = 156), abdominal ultrasound (n = 12) and chest X-ray (n = 12). Overt metastatic disease was reported in five scans (four CT scans and one bone scan). Nonspecific findings (defined as radiologic abnormalities of indeterminate significance requiring follow up as reported by the radiologist) were detected in 89 SIs (68 CT scans, 19 bone scans and two abdominal ultrasounds), which triggered a total of 138 follow-up radiological scans in 71 (42%) of the 168 patients that had initial Sis, including: CT scan (n = 75), ultrasound abdomen (n = 29), MRI liver (n = 14), bone scan (n = 4), chest X-ray (n = 5), bone X-ray (n = 5), MRI spine (n = 4) and PET scan (n = 2). Nine patients underwent biopsy of suspicious lesions. Only one biopsy from mediastinal lymph nodes in a patient with triple negative clinical stage IIB breast cancer was positive for metastatic breast cancer. Early-stage lung adenocarcinoma was discovered in two patients after lung biopsy. Another patient was diagnosed with Gastric Schwanoma after multiple investigations. The remaining five biopsies were negative for malignancy. More details are provided in Tables 2 and 3.

**Table 1.** Demographic and treatment characteristics of all eligible patients.

| Patient Characteristics (n = 295) | N (%) |
|---|---|
| Mean age (years) and range | 63.6 (30–92) |
| Menopausal status | |
| Premenopausal | 55 (19%) |
| Postmenopausal | 240 (81%) |
| Age adjusted Charlson Index | |
| 2–4 | 61 (21%) |
| >4 | 234 (79%) |
| Mode of Diagnosis | |
| Self detected | 151 (51%) |
| Mammogram detected | 144 (49%) |
| T stage | |
| T1a/b | 51 (17%) |
| T1c | 90 (31%) |
| T2 | 154 (52%) |
| N stage | |
| N0 | 215 (73%) |
| N1 | 80 (27%) |
| TNM Stage | |
| Stage I | 123 (42%) |
| Stage IIA | 110 (37%) |
| Stage IIB | 62 (21%) |
| Receptor status | |
| ER and/or PR positive | 249 (84%) |
| HER2 positive | 37 (13%) |
| Triple negative | 36 (12%) |
| Lympho-vascular invasion | |
| Present | 27 (9%) |
| Grade | |
| Grade 1–2 | 197 (67%) |
| Grade 3 | 98 (33%) |
| Surgery | |
| Breast Conserving Surgery (BCS) | 224 (76%) |
| Mastectomy | 71 (24%) |
| Sentinel lymph node biopsy (SLNB) | 276 (94%) |
| Axillary node dissection (AND) | 19 (6%) |
| Systemic therapy | |
| Chemotherapy | 107 (36%) |
| Trastuzumab | 30 (10%) |
| Endocrine therapy | 225 (76%) |
| Radiotherapy | 234 (79%) |

*3.3. Association between Staging, Patient Demographics, Tumour and Treatment Characteristics*

The following patient characteristics were more likely to be associated with baseline SI compared to patients not undergoing SI: younger age (mean age 62 vs. 66, $p = 0.001$), premenopausal status (40 (24%) vs. 15 (12%), $p = 0.009$) and self-detected tumours (101 (60%) vs. 50 (39%), $p < 0.001$). There was no significant difference in mean age adjusted Charlson index between groups.

Pathologic tumour features associated with SI were: T2 stage (122 (73%) vs. 32 (25%), $p < 0.001$), N1 stage (76 (45%) vs. 4 (3%), $p < 0.001$), ER negative status (38 (23%) vs. 8 (6%), $p < 0.001$), HER2-positive status (33 (20%) vs. 4 (3%), $p < 0.001$), triple negative tumours (28 (17%) vs. 8 (6%), $p = 0.007$), grade 3 (80 (48%) vs. 18 (14%), $p < 0.001$) and positive

lympho-vascular invasion (24 (14%) vs. 3 (2%), $p < 0.001$). Mastectomy and axillary node dissection rates as well as the number of patients receiving adjuvant chemotherapy were higher in patients undergoing SI. The mean time between surgery and the initiation of adjuvant systemic therapy was not significantly affected by staging; however, there was a significant delay in starting adjuvant radiation therapy in those with staging investigations (81 versus 148 days, $p < 0.001$). (See Table 4).

**Table 2.** Type and number of baseline and follow-up investigations (n = 295).

| Initial Staging investigations (N = 295) | N (%) |
|---|---|
| Yes | 168 (57%) |
| No | 127 (43%) |
| Frequency of SI | |
| 0 | 127 (43%) |
| 1 | 14 (5%) |
| 2 | 104 (35%) |
| 3 | 38(13%) |
| Total number of SIs | 332 |
| Type of SI | |
| Chest X-ray | 12 (4%) |
| Abdominal ultrasound | 12 (4%) |
| Bone scan | 156 (47%) |
| CT scan | 152 (45%) |
| Follow-up investigations in patients with initial SI (N = 168) | N (%) |
| Yes | 71 (42%) |
| No | 97 (58%) |
| Total number of follow-up investigations | 138 |
| Type of follow-up investigations | |
| X-ray | 10 (7%) |
| Abdominal ultrasound | 29 (21%) |
| Bone scan | 4 (3%) |
| CT scan | 75 (54%) |
| PET scan | 2 (2%) |
| MRI | 18 (13%) |
| Biopsy performed in patients undergoing SI (N = 168) | |
| Yes | 9 (5%) |
| No | 159 (95%) |

**Table 3.** The radiological investigations in the nine patients who underwent interventional biopsy and the results of the biopsy.

| Patient | Radiology Scan Performed | Result of Biopsy |
|---|---|---|
| 1 | CT scan | Lung metastases from breast cancer |
| 2 | CT scan, Abdominal ultrasound | Benign liver lesion |
| 3 | CT scan | Benign lung lesion |
| 4 | CT scan, abdominal ultrasound | Benign adnexal mass |
| 5 | CT scan | Benign lung lesion |
| 6 | CT scan, Abdominal ultrasound, MRI abdomen | Gastric Schawanoma |
| 7 | CT scan, PET scan | Lung adenocarcinoma |
| 8 | CT scan | Insulinoma |
| 9 | CT scan, PET scan | Lung adenocarcinoma |

**Table 4.** The association between baseline staging investigations and patient demographics, and tumour and treatment characteristics.

| | Staging Done (n = 168, 57%) | No Staging (n = 127, 43%) | *p*-Value |
|---|---|---|---|
| Mean age (years) | 62 (SD 12.4) | 66 (SD 11.0) | 0.001 |
| Menopausal status | | | |
| Premenopausal | 40 (24%) | 15 (12%) | 0.01 |
| Postmenopausal | 128 (76%) | 112 (88%) | |
| Mean Age adjusted Charlson | 5.48 (SD 2.17) | 5.11 (SD 1.77) | 0.12 |
| Tumour characteristic | | | |
| T1 | 46 (27%) | 95 (75%) | |
| T2 | 122 (73%) | 32 (25%) | <0.001 |
| N0 | 92 (55%) | 123 (97%) | |
| N1 | 76 (45%) | 4 (3%) | <0.001 |
| ER Negative | 38 (23%) | 8 6%) | <0.001 |
| HER2 positive | 33 (20%) | 4 (3%) | <0.001 |
| Triple Negative | 28 (17%) | 8 (6%) | 0.007 |
| Grade 1–2 | 88 (52%) | 109 (86%) | |
| Grade 3 | 80 (48%) | 18 (14%) | <0.001 |
| LVI present | 24 (14%) | 3 (2%) | <0.001 |
| Surgery | | | |
| BCS | 111 (67%) | 113 (89%) | |
| Mastectomy | 55 (33%) | 16 (11%) | <0.001 |
| SLNB | 153 (91%) | 123 (97%) | |
| AND | 15 (9%) | 4 (3%) | 0.55 |
| Adjuvant chemotherapy | | | |
| No | 74 (44%) | 114 (90%) | |
| Yes | 94 (56%) | 13 (10%) | <0.001 |

## 4. Discussion

Staging investigations for the detection of distant metastases have been part of routine practice for many years, as initial work up of patients with newly diagnosed early breast cancer. There is, however, mounting evidence that routine staging in asymptomatic patients is of low yield and does not alter the course of the disease [14–16].

We undertook a retrospective analysis to evaluate the standard practice at our institution in regard to staging investigations in patients with EBC between January 2012 and December 2014. The main goal was to evaluate how often staging investigations were ordered and what the outcome of the tests was. While we did not formally compare the frequency of staging investigations before and after release of the ASCO recommendations to assess changes in practice over time, we expected a low proportion of staging to be ordered during our study timelines if adherence to the guideline was high. Our findings suggest that the uptake of the guideline in standard practice at our institution was low during the study timelines with more than 50% of patients undergoing unnecessary staging investigations. Unsurprisingly, patients with higher risk disease (larger tumor size, high grade, higher stage and adverse receptor profile) were more likely to undergo staging investigations. Metastatic disease pick-up rate was extremely low and came at a high cost of an unnecessary cascade of follow-up investigations, summative radiation exposure, potentially harmful invasive procedures and patient anxiety [17–19]. Seventy-one of the 168 patients with SI (42%) underwent a total of 138 follow-up imaging tests, and none were ultimately diagnostic for metastases. These results are in keeping with other studies showing similar trends in investigations conducted for follow up of incidental findings [20,21]

Since our study timeline in 2017, Cancer Care Ontario has published and updated their evidence-based practice guideline regarding baseline staging imaging for distant

metastasis in women with stage I, II and III Breast Cancer [12]. For asymptomatic stage I and II breast cancer patients, there is no need for any radiological staging, irrespective of their biomarker profile. On the other hand, all stage III breast cancer patients should undergo baseline staging even in the absence of symptoms. Additionally, the updated 2019 ASCO Choosing Wisely campaign still recommends against using PET, CT or radionuclide bone scans in aEBC [13] We believe that our study together with the existing literature will serve as an important baseline framework for future studies to evaluate changes in practice and adherence to guidelines as well as cost savings to the medical system.

The additional literature was published regarding staging investigations and adherence to guidelines in EBC. Data collected from 25 hospitals in Michigan showed that 20% of 34078 patients diagnosed with stage 0-II breast cancer underwent unnecessary imaging with no evidence of metastatic disease [22]. The authors observed a gradual decline in investigations over time between 2008 and 2015 in EBC, reflecting the uptake of the guidelines into practice and potential cost savings.

A systematic literature review was conducted in Europe to assess the extent of healthcare providers' adherence to breast cancer clinical guidelines and to identify the factors that impact on adherence. The overall breast cancer care process (from diagnosis to follow up) adherence ranged from 54 to 69%. Internal factors that potentially impacted on healthcare providers' adherence were their perceptions, preferences, lack of knowledge, difficulty interpreting guidelines or intentional decisions. External factors included the patient-related factors (such as age and patient's preference) associated most commonly with nonadherence [23] Similar barriers were reported in a systematic meta-review. Some of the most frequent facilitators to implementing guidelines were consistent leadership, commitment of the members of the team, existence of multidisciplinary teams and education regarding the guidelines [24]

Another study conducted at the University of Alabama at Birmingham Health System Cancer Community Network showed that concordance varied according to the Choosing Wisely recommendation, ranging from 39% to 94%. There was significant variability in concordance across centers, with as much as an 89% difference in concordance rates between cancer centers. Non-concordance was associated with higher costs for every measure [25].

Our study has a number of limitations, inherent to a retrospective chart review study (missing charts, incomplete data, single data abstractor and lack of blinding). Furthermore, the study timelines were very close to the original 2012 ASCO Choosing Wisely publication, and we did not formally evaluate if the rate of staging investigations decreased subsequent in our study period. The actual cost of various testing modalities was not evaluated in our study but could be easily deducted from the cost of individual tests performed. The cost effectiveness of SI has been previously reported to be low in other studies. For example, a report in patients with EBC showed that 66 out of 95 patients with clinical stage I/II breast cancer had negative staging scans with a total cost of staging scans estimated at about USD 5720 per patient [26].

In conclusion, during the study period at our institution, more than 50% of patients with EBC underwent staging investigations with low diagnostic yield and triggered a significant number of follow-up investigations. Our results are in keeping with the existing literature in that the detection rate of metastases in aEBC patients is very low and leads to unnecessary potentially harmful investigations. We agree that routine staging in asymptomatic patients with EBC should not be performed routinely. Furthermore, the results suggest that the adoption of the Choosing Wisely recommendation was slow in the first two years after publication. Future studies could address the incorporation of the guidelines into standard practice over time.

**Author Contributions:** D.K.—methodology, data curation and collection, writing original draft preparation, writing—review and editing, review and editing, approval of final version; V.Y.—writing original draft preparation, review and editing, approval of final version; W.M.H.—methodology, formal analysis and interpretation, assisting with manuscript development, and approval of the final version; M.M.—conceptualization; data analysis and interpretation, writing—review and editing, approval of final version. All authors have read and agreed to the published version of the manuscript.

**Funding:** This research received no external funding.

**Institutional Review Board Statement:** The study was conducted according to the guidelines of the Declaration of Helsinki, and approved by the Institutional Review Ethics Board of Queen's University (File # 6015517; date of approval 21 May 2015).

**Informed Consent Statement:** Patient consent was waived due to retrospective chart review study with no associated risks to participants.

**Data Availability Statement:** The data presented in this study are available on request from the corresponding author. The data are not publicly available due to ethics approval.

**Conflicts of Interest:** The authors declare no conflict of interest.

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
