# Peer review of "Staging Investigations in Asymptomatic Early Breast Cancer Patients at the Cancer Centre of Southeastern Ontario"

_curroncol, doi:10.3390/curroncol28030203_

Round 1

Reviewer 1 Report

The authors describe a single-institution retrospective chart review that occurred following a change in staging guidelines for early breast cancer. The manuscript addresses an interesting topic but is significantly outdated.  Further, it is unclear what purpose it would serve currently in the published literature.

Major Limitations:

  1. A major limitation is that the review is outdated, from 2012 – 2014. In fact, the authors state that guidelines have been updated since that time in 2017 with new recommendations.
  2. Notwithstanding the above, I am unclear on the purpose of the study. Was there uncertainty in the guideline, thus requiring confirmatory evidence?  Were the authors trying to understand the reasons why guidelines were not implemented properly?  If so, there should be more information on this, as it is not clear that this has been explored.
  3. A large part of the discussion section is repetitive to the introduction (first three paragraphs) or seemingly unrelated (section on oligometastatic disease).

Minor Limitations:

Abstract:

  1. In results section, it would help to state that you identified 295 eligible asymptomatic Further, what is the denominator for the 71 patients such that the percentage was 24%?

Introduction:

  1. Line 93-95 – sentence structure is off.
  2. Line100-102 – a citation for Choosing Wisely and a very brief description of its overall goal (not simply the ASCO goal) would be helpful.
  3. Line 113-114 – as in major limitations #2, the authors should include why this study is important and/or relevant.

Methods:

  1. Line 119 – CCSEO is not defined.
  2. Line 119-122 – data presented here should go into results section. The methods section should simply define inclusion/ exclusion criteria and methods.  In addition, it would help to understand how you defined ‘symptomatic disease’ from the chart review.  Was a second extractor used?  Were there discrepancies? How were these resolved?

Results:

  1. Line 141-145 – this part could be shortened and readers referred to the table.
  2. Lines 166-177 – when giving the differences, please give percentages or differences and p-values - - the absolute number is not meaningful to the reader.
  3. Was there any consideration for a multivariable model given the number of statistically significant associations?
  4. Was the type of provider (surgeon, medical oncologist) who ordered the staging investigated?

Discussion:

  1. The first three paragraphs are essentially a repetition of the introduction and should be removed.
  2. One could add a paragraph discussion the associations found to be significant and perhaps tie this to other existing literature.
  3. It is unclear how the section on oligometastatic disease is relevant. The authors have not provided data regarding oligometastatic disease in the results section.  Further, it’s unclear how this is related to baseline staging or the manuscript in total.  If the authors feel it is, this should be explained.
  4. Some discussion on why the authors hypothesize (from their data or the literature) why staging is still being performed, barriers to changing practice, or future directions could provide more substance to the discussion.  
  5. Limitations section?

Author Response

Many thanks for taking the time to review this manuscript. We had reviewed the manuscript thoroughly and had addressed all your recommendations. Please find attached the reviewed manuscript and the reply to all the points in line. Thank you

RESPONSE TO REVIEWER 1 

Comments and Suggestions for Authors

The authors describe a single-institution retrospective chart review that occurred following a change in staging guidelines for early breast cancer. The manuscript addresses an interesting topic but is significantly outdated.  Further, it is unclear what purpose it would serve currently in the published literature.

Major Limitations:

  1. A major limitation is that the review is outdated, from 2012 – 2014. In fact, the authors state that guidelines have been updated since that time in 2017 with new recommendations.

We agree that the study timelines are outdated but we believe the results are still relevant as more recent publications indicate that staging in early breast cancer is still performed frequently and adherence to guidelines is very variable. Our study can serve as a benchmark for future comparisons as to adoption and adherence to guidelines. Additionally, we are also reporting the number of additional investigations triggered by staging investigations that were performed unnecessarily. 

  1. Notwithstanding the above, I am unclear on the purpose of the study. Was there uncertainty in the guideline, thus requiring confirmatory evidence?  Were the authors trying to understand the reasons why guidelines were not implemented properly?  If so, there should be more information on this, as it is not clear that this has been explored.

Indeed the purpose of our study was to evaluate the frequency of staging investigations at our institution as there was growing evidence in the literature as to the low metastatic pick-up rate in asymptomatic patients with early breast cancer. We did not formally assess adoption of the guideline by comparing rates before and after release of the Choosing Wisely recommendations in 2012. Our findings do suggest that adoption was low because of the high rate of performed staging investigations. We have made corrections in the manuscript to clarify the objective of the study.

  1. A large part of the discussion section is repetitive to the introduction (first three paragraphs) or seemingly unrelated (section on oligometastatic disease).

Discussion section was revised. We agree that the section on oligometastatic disease is unrelated and this was removed. 

Minor Limitations:

Abstract:

  1. In results section, it would help to state that you identified 295 eligible asymptomatic Further, what is the denominator for the 71 patients such that the percentage was 24%?

Asymptomatic was added in the result section. The denominator for the 71 patients (24%) is the total number of patients (295).

Introduction:

  1. Line 93-95 – sentence structure is off.

We agree and have removed this sentence.

  1. Line100-102 – a citation for Choosing Wisely and a very brief description of its overall goal (not simply the ASCO goal) would be helpful.

Revised

  1. Line 113-114 – as in major limitations #2, the authors should include why this study is important and/or relevant.

As per response in major limitation #2 we have revised line 113-114.

Methods:

  1. Line 119 – CCSEO is not defined.

Corrected.

  1. Line 119-122 – data presented here should go into results section. The methods section should simply define inclusion/ exclusion criteria and methods.  In addition, it would help to understand how you defined ‘symptomatic disease’ from the chart review.  Was a second extractor used?  Were there discrepancies? How were these resolved?

We have corrected part of this section as suggested but retained the description of data collection. We relied on the patient chart for definition of symptomatic status, specifically patients who had negative review of systems, with no symptoms of metastatic disease, were considered asymptomatic from the breast cancer point of view. There was not a second extractor used for data collection therefore we cannot comment on discrepancies/resolution thereof.

Results:

  1. Line 141-145 – this part could be shortened and readers referred to the table.

Revised

  1. Lines 166-177 – when giving the differences, please give percentages or differences and p-values - - the absolute number is not meaningful to the reader.

This is a good point, and the percentages and p-values have been added to the manuscript, results section.

  1. Was there any consideration for a multivariable model given the number of statistically significant associations?

There are a number of statistically significant associations, but it was not our intent to try to predict whether or not the patients would receive staging investigations.  So it’s not clear what a multivariable model would add to the manuscript. The other challenge is that there is likely to be a high degree of colinearity between the predictors – for example age and menopause status – which would limit our ability to assess these individual factors.  We did make an attempt but the model was unstable, with colinearity a particular problem for ER and Triple Negative, and an odds ratio in the millions for the constant as a consequence.  So we believe that the current bivariate analysis is the most informative.

  1. Was the type of provider (surgeon, medical oncologist) who ordered the staging investigated?

We did not collect this information.  

Discussion:

  1. The first three paragraphs are essentially a repetition of the introduction and should be removed.

Agree. Replaced with suggestion in #15

  1. One could add a paragraph discussion the associations found to be significant and perhaps tie this to other existing literature.

Revised

  1. It is unclear how the section on oligometastatic disease is relevant. The authors have not provided data regarding oligometastatic disease in the results section.  Further, it’s unclear how this is related to baseline staging or the manuscript in total.  If the authors feel it is, this should be explained.

We agree this has been removed.

  1. Some discussion on why the authors hypothesize (from their data or the literature) why staging is still being performed, barriers to changing practice, or future directions could provide more substance to the discussion.  

We agree and had added in a paragraph in the discussion section with new reference

  1. Limitations section?

Added limitation section.

Reviewer 2 Report

This is an interesting retrospective, single center cohort study that seeks to evaluate the use of staging investigations among patients with T1/T2 N0-1 early breast cancer who do not have symptoms of metastatic disease. A cohort of patients treated at the CCSEO between Jan 2012 - Dec 2014 was selected to determine uptake of the 2012 ASCO Choosing Wisely Campaign, which aimed to discourage the use of routine staging investigations among patients with low risk early breast cancer.

Major Comments

1) Cancer Care Ontario Guidelines provide imaging recommendations by cancer stage (I, II, III). Why were patients with stage IIB disease (T3NO) not included in this study? Also it would be helpful to display results by stage (stage I vs stage II) disease as opposed to by the presence of T1-2 disease and N0-1 disease separately.

2) In the Introduction section of the manuscript, the first two paragraphs indicate that 18% of patients with stage I and 41% of patients with stage II cancers had positive bone scans. But "since then, ….yield of staging tests is much lower". This needs to be explained. Why is there a discrepancy between these findings and current recommendations not to screen?

3) When in 2012 were the ASCO Choosing Wisely recommendations published? It is noted that data collection for this study started in early 2012 - does this truly reflect practice patterns after publication of these recommendations?

4)  Given that there is often a delay in uptake of published guidelines/recommendations, it would be interesting to see if the use of staging imaging declined over the course of the 3-year study period.

5) In the Results section, comparisons of groups that did vs did not receive staging investigations should be reported as proportions (not absolute numbers).

6) Were patients treated with neoadjuvant chemotherapy more likely to have staging investigations?

Minor Comments

1) The Results section of the abstract was difficult to follow at times. I would recommend that results be summarized in full sentences, incorporating the most meaningful findings.

2) The sentence describing ref #8 (introduction) is confusing; this should be re-worded. I would also omit reference to surveillance MRI of the liver in the introduction for clarity/flow.

3) Relevant, updated CCO guidelines should be mentioned in the introduction section.

4) Desfine CCSEO in the Methods section.

5) How were non-specific findings defined on baseline staging investigations? Did those staging investigations require follow-up imaging by definition?

6) In Results section, the % of patients with HR+/HER2-, TNBC and HER2+ disease should be clearly outlined in the text.

Author Response

Many thanks for taking the time to review this manuscript. We had reviewed the manuscript thoroughly and had addressed all your recommendations. Please find attached the reviewed manuscript and the reply to all the points in blue. Thank you

Reviewer 3 Report

This is an interesting analysis of staging investigations performed at a single institution for asymptomatic EBC patients. The topic is of relevance, methods are appropriate and no language improvement is required.

In the opinion of this reviewer, the following revisions could benefit the manuscript in view of a publication on this Journal: 

  • In the Abstract, Results could be better described, particularly regarding the description of the population, which currently appears as a long, unnecessary list of terms and percentages
  • In the manuscript, when describing the Results, several statistical differences are infered, but no p-value neither any other statistical metric is explicited. I would add statistical results into the manuscript, for a better interpretations of the study outcomes
  • More synthesis and removal of repetitions would benefit the Discussion. For instance, "routine staging in asymptomatic patients is of low yield" is repeted twice, as well as "Routine imaging can lead to harm, radiation exposure....". 
  • In the Discussion, authors state that "The main goal was to evaluate if standard practice had changed around the time of the ASCO choosing wisely guideline publication". However, the study does not appear to be designed to assess this endpoint, whereas it simply describes the staging investigations in a cohort of patients according to their characteristics. This measleding description of the outcome should be removed.

Author Response

(The authors gave the same response as above.)

Round 2

Reviewer 1 Report

The authors describe a single-institution retrospective chart review that occurred following a change in staging guidelines for early breast cancer. The manuscript addresses an interesting topic, although given reference to previous systematic reviews on the topic, not a particularly novel one.  The included objectives are now somewhat more clear to the reader.  A few additional limitations remain.

Minor Limitations:

Abstract:

  1. The authors responded that the denominator for the 71 patients that underwent follow-up tests is the total number of patients (295). It would be helpful to clarify this by putting the denominator in the abstract, as technically, a patient is only ‘eligible’ to receive a follow-up test if she had baseline SI in the first place (ie the true eligible denominator is the 168 patients who received staging).  If the authors would like to use the total number this should be clarified - - this is important as you have used this to compare to other studies as well.
  2. Further, this result appears in the abstract and discussion section, but not in the results section of the manuscript. If important enough for these other locations, it should be in the main results section of the paper.

Methods:

  1. Thank you for clarifying your extraction methods and definition of symptomatic status. If space allows, this would be important to include in the manuscript.

Discussion:

  1. The citations at the end of the 2nd paragraph of the discussion (citations 17,18) appear to be about the risks of radiation, not other studies showing similar trends for follow-up of incidental findings. Are these meant for a different part of the discussion?

Table 2:

Similar to the abstract, the denominator/percentages are difficult to follow in Table 2 – for example, “type of SI” seems to also use 295 as the denominator (ie 295 eligible participants) but I would expect 332 (total # of SI) to be used as the denominator (when in fact if you add up everything in the column it is - - evidenced by the fact that the percentages add up to over 100%)

Second, the categories switch people total # of patients and total # of scans, which is confusing to follow.  (Ie I believe the row labeled “follow-up investigations” is actually # of patients with follow-up investigations….). 

Therefore, I recommend that each row be clearly labelled in regards to a) is the category referring to # of patients of # of scans and b) what exactly the denominator is (total population, only the population who received SI, etc) and its value.

On this note, in the results section, it would also help to clarify # of patients vs # of scans, where relevant.

Author Response

Many thanks for taking your time to review. We had adjusted the manuscript as per your recommendations. Thank you again for all your review that had enriched the manuscript.

Reviewer 2 Report

The revised manuscript is now acceptable for publication.

Author Response

Many thanks for taking time to review the manuscript. Your review that had enriched the manuscript. 

Reviewer 3 Report

This reviewer is satisfied with the revisions.

Author Response

(The authors gave the same response as above.)
